# Interventions Which Aim at Implementing the Knowledge-Based Approach in the PE Lesson: A Systematic Review

Teodora Mihaela Iconomescu [1,*], Bogdan Sorin Olaru [1], Laurențiu Gabriel Talaghir [1,2], Claudiu Mereuță [1], Gheorghe Balint [3], Marian Florin Buhociu [4,*] and Viorel Dorgan [5]

1    Faculty of Physical Education and Sport, Dunărea de Jos University, 800008 Galati, Romania; bogdan.olaru@ugal.ro (B.S.O.); gtalaghir@ugal.ro (L.G.T.); cmereuta@ugal.ro (C.M.)
2    Institute of Sport, Tourism and Service, South Ural State University, 454080 Chelyabinsk, Russia
3    Faculty of Movement, Sport and Health Sciences, Vasile Alecsandri University, 600115 Bacau, Romania; gbalint@ub.ro
4    Doctoral School of Fundamental Sciences and Engineering, Dunărea de Jos University, 800008 Galati, Romania
5    Cross-border Faculty, Dunărea de Jos University, 800008 Galati, Romania; dorganv@gmail.com
*    Correspondence: ticonomescu@ugal.ro (T.M.I.); fbuhociu@ugal.ro (M.F.B.)

**Abstract:** Background: Lately, there has been a change in the approach to physical education as a school subject. The new (knowledge-based) approach proposes the teaching of a theoretical component that provides information and complements the practical one. The students thus acquire a thorough understanding of the principles underlying physical activity and assimilate the knowledge needed to independently conduct their physical activity throughout their lives. Materials & Methods: Firstly, there were identified a number of interventions that implement the theoretical component specific to the new approach, in the school environment. Interventions targeting students from the first grade to the university level were taken into account, without setting any geographical or temporary limits. Then, we analyzed the way in which the theoretical content was adapted and implemented at each educational level. The tools used in the evaluation of the theoretical component were also presented. Results and Conclusion: Even from the elementary school level, we find adapted methods for implementing a cognitive component. As we advance through middle school, high school and university level, we find interventions that propose theoretical contents adapted to contemporary society. Within the university-level chapter, special attention was dedicated to future physical education teachers and to the way in which they are prepared to teach a cognitive component within the physical education lesson. Finally, three categories of tools used in the evaluation of the cognitive component were presented: questionnaires, interviews, interactive methods.

**Keywords:** physical literature; physical education; knowledge-based approach; school interventions

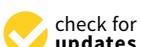



## 1. Introduction

The historian Park (1989) [1] studied the transformations that the school subject of Physical Education went through over the years. The author asserts that the first crystallization movements took place around 1800, the ideal towards which this school subject aspired at that time being very close to the military environment—with well-established exercises combined with gymnastic exercises. A change in this ideal was identified at the beginning of the 20th century as a result of the "democratic values". This period was characterized by sports games that promoted values such as leadership, teamwork and fair play. Following this "sport-based approach", Park (1989) [1] predicted that the emergence of a new science in physical education—kinesiology—would be the catalyst that would transform PE again, in the 21st century, with this approach relying on the implementation of a theoretical content that would provide the student with an in-depth understanding of physical activity (knowledge-based approach).

Twenty years have passed since we entered the 21st century and it is becoming more obvious that the ideal of Physical Education is changing. Hargreaves (2003) [2] refers to the 21st-century society as the "knowledge society", in which students are encouraged to develop critical thinking. In this environment, Physical Education had to adapt to the new contemporary ideal. Thus, the present-day approach to PE highlights the importance of a theoretical component, in addition to the practical one, giving the latter much more sense.

Being a new approach, the need for this systematic review is obvious. The present study presents, on the one hand, the concrete ways in which this "knowledge-based approach" is implemented and evaluated at different age levels and, on the other hand, the way in which future teachers are prepared to teach this theoretical content. In order to achieve this, we consider it necessary to delimit the role of the cognitive component in physical education before presenting concrete ways of implementing and evaluating this approach.

### 1.1. Physical Education in Postmodernism

Revitalizing the question asked by Young (2008) [3], Ennis (2015) [4] applies it to PE and asks: "What is my responsibility to teach my students?". The answer revolves around two elements: knowledge and skills. From this perspective, the author notes that a physically educated individual involves an individual with a thorough understanding of basic concepts, principles and procedures. This individual should not only possesses theoretical knowledge but also knows how to apply this knowledge [4].

Fullan (2001) [5] emphasizes the fact that current education focuses on what students know, namely declarative knowledge, and on what they can do-applicative knowledge-in relation to the specifics of the school subject. This approach, applied to PE, requires that students be evaluated, on the one hand, according to their knowledge and understanding and, on the other hand, according to their ability to apply this knowledge-not according to a standardized performance scale. On the same topic of shifting the focus from the evaluation of motor performance, Jackson (2006) [6] wrote a paper in which he follows the evolution of fitness testing from the period with a strong athletic emphasis to the contemporary period, characterized by public health. The author also analyses the forces that bring about this change in an environment that is not exactly receptive. This view is reinforced by Macdonald (2011) [7], who states that testing and reporting motor performance tests or body mass index may be contrary to the educational intent of the school subject.

What we want to emphasize is that contemporary physical education has shifted its focus from the sport-based approach to the approach in which the promotion of an active and healthy life is in the foreground.

### 1.2. Physical Literacy (PL)

The term physical literacy (PL) was proposed for the first time in 1993 by Margaret Whitehead at the International Association of Physical Education and Sport for Girls and Women Congress in Melbourne, Australia [8]. In this first phase of crystallization, it was defined as a holistic approach, which encompasses knowledge, skills and motivation, with the help of which an individual engages in physical activity for life [9].

Once proposed, the term enjoyed international success and was adopted by many countries to define the ideal citizen from the perspective of physical education. It is true that each country has nuanced it by emphasizing one area or another, but the essence of the multidimensional approach and the focus on health promotion have been preserved [10]. In Canada, for example, in order to be able to implement this vision successfully, the need for a formal and clear definition of the term was pointed out, but also a series of clarifications and guidelines for practitioners. To this end, the Vancouver Declaration of the International Physical Literacy Conference approved the definition of physical literacy as: "the motivation, confidence, physical competence, knowledge and understanding to value and take responsibility for engagement in physical activities for life" ([9] p. 16).

As it can be seen from the above definition, there are several forces that may act on the individual (motivation, confidence, physical competence, knowledge and understanding).

In Australia, the document entitled "Physical Literacy Framework" presents the concept of "physical literacy" as being made up of four areas of interest: (1) physical, (2) psychological, (3) cognitive and (4) social [11].

From these two examples, it can be seen that the differences in terminology are minor. The same conclusion was reached by Cornish et al. (2020) [12], who looked at different approaches to the term physical literacy. His conclusion indicates the existence of 4 seemingly distinct domains that make up physical literacy: (1) the emotional domain, (2) the physical domain, (3) the cognitive domain, and (4) the behavioural domain. The emotional domain refers to motivation and confidence, the physical domain to physical capacity, the cognitive domain refers to theoretical knowledge and its understanding, and the behavioural domain refers to involvement in physical activities all throughout life [12].

Physical literacy is not a finish line; it is a journey that humans make all throughout life. But in order to embark on this path, students have to learn how to learn during school years [2]. They must also have self-confidence and self-esteem to be motivated to continue on this path, and to do so, the teacher must value any progress made by any student without comparing motor performance with a standardized scale [10]. Ennis (2015) [4] believes that the gateway to the path of physical literacy is represented by theoretical knowledge—by the acquisition, understanding and ability to practically apply the theoretical content. The same opinion is shared by Corbin (2020) [13], who considers that the implementation of theoretical content is an important feature of physical literacy. The author continues by encouraging physical education teachers to pay more attention to the theoretical content in the teaching process.

Young et al. (2020) [14] draw attention to the danger of PL being viewed separately through one or more of the four areas that make it up. However, the authors go on to say that any of the four dimensions can be addressed separately for research or evaluation purposes only. Thus, we will try to extract the cognitive component for analysis that is said to be the first step towards physical literacy.

*1.3. The Conceptual Physical Education (CPE)*

Referring to the cognitive component, Sport Australia (2009) [11] describes it as "A person's understanding of how, why and when they move". This understanding is obtained through seven elements: "(1) Content knowledge; (2) Safety & risk; (3) Rules; (4) Reasoning; (5) Strategy & planning; (6) Tactics; (7) Perceptual awareness" ([11] p. 8).

The term conceptual physical education (CPE) is nothing more than the cognitive component of physical literacy. Corbin & Laurie (1978) [15] define it as a physical education program that focuses on teaching and understanding the concepts, principles, and techniques of independent physical activity management, with the goal of promoting a healthy lifestyle and its outcomes. The distinctive feature of conceptual physical education is that, in addition to traditional PE lessons, theoretical lessons are used that take place in the classroom, in which a textbook or other printed material is used [15].

Corbin (2020) [13] says about conceptual physical education that it is the activity in which the student first acquires theoretical content in the classroom, using a printed textbook/material, and then participates in practical lessons designed specifically to apply the previously acquired theoretical content. Placek et al. (2001) [16] point out that the lack of theoretical knowledge on which to base a healthy and active life is as worrying as misunderstanding the content. The authors emphasize the need for the teacher to ensure not only the acquisition of theoretical knowledge but also the correct understanding of that knowledge.

The beginnings of this theoretical approach were meant to complete the practical one and were found at the university level where it appeared under various names: Fitness for Life, Personal Fitness, Concepts of Fitness and Wellness, etc. However, after the demonstrated success at this level, it was quickly adopted at the other levels of education [13].

In the present paper, we will try to provide an overview of how this cognitive component can be addressed in the physical education lesson. To begin with, we will analyse the intervention programs at the level of the education system that is aimed at implementing a theoretical component in PE. After that, we would like to extract from these interventions and order, by age levels-the main methods, means, and contents of the cognitive component of physical literacy. Finally, we will try to present the evaluation tools used by the intervention programs identified by us.

## 2. Materials and Methods

The identification of the relevant works took place between 1 July 2020 and 1 June 2021. For this, the SCILIT database was used, developed, and maintained by MDPI. This database offers the advantage that it imports the scientific material indexed in all the important specialized journals (319 journals) and all the material from the CrossRef and PubMed databases on a daily basis. This means that, by a single search in this database, the cumulated results from all representative databases are obtained.

To identify the papers, combinations of the following keywords were used: physical literacy/conceptual physical education cognitive component/theoretical component/physical education/intervention/program/pupils/students/teachers/pre-service teachers/school.

In order to be selected, the papers had to cumulatively meet the following requirements: (1) to be an intervention or a program that implements the cognitive component specific to the new approach (knowledge-based approach) to PE; (2) the intervention had to be performed in a school-setting; (3) the intervention had to address the educational levels between the first grade and the university level.

No limits were established in terms of any limitation regarding the year of publication. This fact allowed us to establish both the period of the first attempts to implement theoretical content and the period during which most implementation programs were conducted. No geographical limitations were set, with the only condition in this regard being that the paper is written in English.

As a result of the search, 31 interventions have been identified that meet the established criteria. Figure 1 illustrates the process by means of which these interventions were selected.

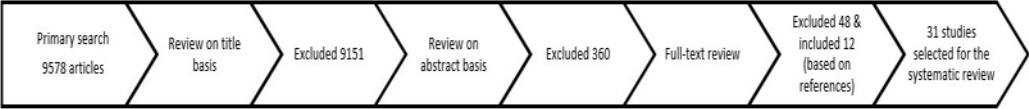

**Figure 1.** The process of selecting the works.

The remaining 31 interventions were then grouped on four educational levels: (1) elementary school (approx. 6–10 years of age); (2) middle school (approx. 10–14 years of age); (3) high school (approx. 14–18 years of age) and (4) university-level (over 18 years of age). This classification was made in order to exemplify how the cognitive component of physical education was adapted to each educational level.

For each selected intervention, a qualitative analysis was performed, containing the main characteristics of the intervention and the tool used to assess the cognitive component.

After completing these stages, the present paper was written, which presents the most representative ways to implement the cognitive component in the physical education lesson, adapted to each educational level, and the tools used by these interventions for the assessment process.

## 3. Discussion

Before starting the presentation on educational levels, we selected two intervention programs that we cannot integrate into any category because they address several levels.

"The Society of Health and Physical Educators," also known as "SHAPE America", is an organization that deals with the fields of health, physical education, leisure and dance. This organization is also responsible for drafting performance standards, i.e., what

each student should know and what he/she should know how to do after attending the physical education class, depending on his age. Here is how the "National PE Standards" are presented [17]:

- Standard 1: The physically literate individual demonstrates competency in a variety of motor skills and movement patterns.
- Standard 2: The physically literate individual applies knowledge of concepts, principles, strategies and tactics related to movement and performance.
- Standard 3: The physically literate individual demonstrates the knowledge and skills to achieve and maintain a health-enhancing level of physical activity and fitness.
- Standard 4: The physically literate individual exhibits responsible personal and social behavior that respects self and others.
- Standard 5: The physically literate individual recognizes the value of physical activity for health, enjoyment, challenge, self-expression and/or social interaction [17].

As can be seen, these standards use the term "physical literacy," but, as Corbin (2020) [13] explains, in the United States, even if the definition proposed by the International Physical Literacy Association (2016) is accepted, a physically literate person is considered the person that meets National PE Standards. These standards represent the first step in transposing theory into practice-the next step is represented by the concrete examples of content that each physical education teacher receives to meet these standards at each age level. In an attempt to provide as clear a description as possible of how the cognitive component is implemented, we will cite the content of Standard 4, which a California PE teacher receives in the guide. (Table 1) We will cite both the contents for grade one and for the 8th grade in order to underline the progression in the complexity of the proposed theoretical notions.

**Table 1.** Performance standard 4: grade one vs. grade eight.

| Standard 4 Students Demonstrate Knowledge of Physical Fitness Concepts, Principles, and Strategies to Improve Health and Performance | |
| --- | --- |
| **Grade One** | **Grade Eight** |
| Similar to the relationship between Standards 1 and 2, Standard 4 provides the cognitive information to support the fitness activities experienced in Standard 3. Specifically, students learn about physical activities that are enjoyable and challenging, the names of internal parts of the body (e.g., bones, organs), how muscles are used for climbing and for moving bones, and the need to stretch muscles to keep them healthy. They also learn that the heart is a muscle, and it works with the lungs to send oxygen to the other muscles throughout the body. Kindergarten students learn the role of nutrition (including the importance of water) in providing energy for physical activity ([18] p. 27). | Similar to the relationship between Standards 1 and 2, Standard 4 provides the cognitive information to support the fitness activities described in Standard 3. For Standard 4, the students refine their fitness plans. Eighth-grade students are building upon their seventh-grade experience in creating a personal fitness plan by expanding it from one to two weeks. This experience is preparation for developing fitness plans throughout their lives. Students also identify appropriate substitute physical activities for times when their usual fitness program is disrupted by inclement weather, travel, or minor injury. Eighth-graders explain different types of conditioning to support different physical activities. They identify safety procedures for and apply basic principles in resistance training activities. They are also able to explain how nutrition and participation in physical activity impact weight control, self-concept, and physical performance. This is a crucial time to help students apply their knowledge, so they can use it for the remainder of their lives ([18] p. 105,106). |

From this example, it can be seen that the theoretical knowledge is designed to be implemented progressively, starting with basic notions and ending with strategies for applying the principles and concepts that were previously taught.

Another approach we have chosen to present here is the one proposed by Viggiano et al. (2014) [19], who developed a board game with which he obtained remarkable results because it addresses a wide range of ages (9–19 years). The game is an innovative method, highly appreciated by students of all ages. After six months of implementation, improvements in the level of nutritional knowledge and related to physical activity were reported, as well as positive changes in eating habits. In addition, 18 months after implementation, there have been improvements in BMI.

These two programs are clear examples that demonstrate that the theoretical component has already been widely implemented with outstanding results at all levels of education.

The fact that in the United States, we find standard 4, which focuses strictly on theoretical knowledge, indicates that this (knowledge-based) approach no longer needs to demonstrate its benefits, as long as it has already been introduced in mainstream education.

*3.1. Elementary School (about 6–10/11 Years of Age)*

The way in which theoretical knowledge is transmitted to students is an essential factor in ensuring success, regardless of the educational level we are talking about. But, as Sun et al. (2012) [20] point out when we talk about the level of elementary education, the teacher must adapt the form of presentation of the content to the understanding possibilities of the team.

We will start presenting the cognitive interventions identified by us, at this age, with two programs based on the constructivist approach that involves "The 5 Es learning cycle lesson structure".

The SPEM curriculum (Science, PE, & Me) was implemented in America between 2003 and 2008, for the third, fourth, and fifth grades, and had as a basic principle the creation of a meaningful progression in which students acquired knowledge and understood it. In this curriculum, students study both the components of fitness and how to apply fitness concepts, principles and procedures related to physical activity, nutrition and health. Each lesson has been designed to contain the 5 Es, as follows: (1) Engagement—with the help of games or physical challenges, concepts or principles that underlie these activities are introduced; (2) Exploration/Experiment comprises 65–75% of the total time allocated to the lesson, and here the students make the connection between the proposed concept and its manifestation in practice. For example, they learn how exercising increases the heart rate or how lifting weights or performing an exercise for a certain number of times affect the muscle; (3) Explanation—this is the cool-down stage, after the effort, in which the students discuss among themselves certain questions asked by the teacher; (4) Elaboration—relational links are created between the concept/principle approached and previous knowledge and the students learn how to use the content taught in their free time, at home or in performance sports; (5) Evaluation—in this last stage, the students answer short questions or fill in data in the "Personal Scientific Journals" [4].

The second program was created by Sun et al. (2012) [20]. Here, the experimental group focuses on the cognitive side, and students are turned into "scientists"—they will have to keep a diary in which to make predictions, write down observations and collect data on how their body reacts to physical activity, then the teacher conducts debates and discussions based on the information from students' diaries. Students must also consider the implications of their "discoveries" in everyday life and discuss with others to clearly establish the place of this information in the social context. One of the conclusions drawn from this experiment is that such an approach is in line with the visions of educational psychology that state that man controls his behaviour by applying the knowledge stored in the cognitive system. In the program proposed in this experiment, students are asked exactly this, to make the connection between physical activity and the previously acquired theoretical knowledge [20].

Three other identified research studies, which we chose to present, have in common the involvement of parents in the process of learning and understanding the theoretical component for the formation of healthy habits.

An interesting approach is that of the "Family Fitness" program [21], which seeks to provide theoretical knowledge to both third-graders and their parents. Starting from the idea that the goal of any health program is to educate children, the authors identify their parents as the main actors that can promote and influence the behaviour of primary school students. The applied intervention involves a complex of actions: courses for teachers who will deliver the content, courses for parents, courses for students and last but not least, the actual development of the program. The latter involves a package of information, exercises and food tasks, which each child brings home every Monday. By the end of the week, students, together with their parents, must obtain as many points as possible by completing the tasks included in the "weekly package".

Hopper et al. (1996) [22] also identified the importance of parental involvement at this level, in an intervention that involved a combination of two elements: (1) physical activities for students; (2) teaching food notions to both students and their parents.

The third identified program, which also presupposed the involvement of parents in the training/learning process, is "Be Smart". This program highlighted, first of all, the individual results offered by an intervention based only on nutritional knowledge (Eat Smart) and another one based only on physical activity (Play Smart). After that, another third group was subjected to both intervention programs (Eat Smart, Play Smart) and thus demonstrated the importance of a mix of theory and practice [23].

The following three intervention programs are characterized by innovative approaches to delivering theoretical content for this age group.

Caballero (2003) [24] used as a tool to implement theoretical knowledge, a classroom course in which students listened to stories with imaginary characters in which the heroes were on a journey to "a healthy lifestyle." The program, called "Pathways", focuses on 4 components: (1) changes in diet, (2) increasing the level of physical activity, (3) a curriculum for a theoretical approach to a healthy lifestyle, (4) a program for family involvement.

"MyPlate the Musical" is another program that uses as a method, a staging by students of a musical, with a subject related to nutrition and physical activity. The results indicated improvements in the level of theoretical knowledge both in the students who participated as actors and in the spectators—with the mention that the "actors" acquired more knowledge than the "spectators" [25].

The KOPS (Kiel Obesity Prevention Study) program aimed at delivering the messages: "eat fruits and vegetables every day", "reduce the consumption of fatty foods", "exercise one hour a day" and "reduce the time spent on TV at an hour a day". These messages were delivered, adapted to the age, in the form of stories, interactive games and by preparing healthy meals. Six such lessons were offered to students for about three weeks, and each lesson was followed by 20 min of physical activity in the schoolyard [26,27].

The last two intervention programs, selected for presentation at this level, involve obvious theoretical approaches.

One of the most important programs implemented in the United States is "Know Your Body" [28]. It was developed in 1970 by the American Health Foundation (AHF) and had, as an experimental group, pupils from the first to the 6th grade. The goal is to equip students with attitudes, knowledge, skills, and experiences necessary to form healthy behaviours. At the beginning of each of the six years in which the program takes place, students receive books and/or worksheets. In addition, each teacher receives a guide to help him implement the program. In order to go through the theoretical content of the curriculum, 40 lessons are scheduled to take place in the classroom. Another component of the program is the "mirror of health" of each student. This involves measuring: (1) height/body weight, (2) blood cholesterol, (3) blood pressure, (4) fitness tests. This data is then used in the theoretical classes so that the students can make the connection between the acquired knowledge and the implication that this knowledge has in practice. Parents receive letters about

the student's situation and there are activities in which the parents are directly involved. Another feature of the program that should be mentioned is that evaluation campaigns are carried out periodically at a national level in order to be able to report the quantity and the degree of understanding of the notions assimilated by the students [28].

Knisel et al. (2020) [29] implemented a program that aimed at developing theoretical knowledge, practical applications, critical thinking and self-awareness in students aged 6 to 12. To do this, he designed an intervention program in which students went through each piece of information in four methodical steps: (1) searching for and finding relevant health-related information; (2) understanding the information; (3) interpreting and evaluating information; (4) sharing and using the information to improve health.

One may easily notice that the implementation of theoretical knowledge at this educational level is possible, and the methods used in the interventions presented above are extremely ingeniously designed to adapt to the particularities of elementary education.

The theoretical notions that these interventions aim at implementing touch on a wide range of topics, which indicates the fact that this educational level allows the introduction of a theoretical component that complements the practical component. In this way, physical activity acquires much more meaning for the student, and all this knowledge, if systematically introduced, can be a solid basis for practising independent physical activity throughout one's life.

### 3.2. Middle School (about 10–14 Years of Age)

The philosophy behind the notion of "physical literacy" has had an innovative approach in the SHL (The Science of Healthful Living) program developed by the National Institutes of Health as a result of the success of SPEM program (Science, PE, & Me)-intended for the primary cycle [4]. SHL targeted sixth, seventh and eighth-grade students and the main objective was to expand students' knowledge and understanding about the cardio-respiratory system and nutrition. In addition to these two main objectives, the curriculum, which was developed between 2011 and 2016, included content related to scientific concepts of fitness and health, such as stress management, the influence of media content on health and goal setting. The SHL program includes 120 lessons divided into two units distributed each year of study: "The Cardio Fitness Club" and "Healthy Lifestyles". In the first year (sixth grade) a theoretical knowledge base is established, introducing essential fitness concepts, such as: FITT (frequency, intensity, type and time), progressive increase of effort, energy production for aerobic effort and anaerobic, calories, caloric intake, caloric balance, etc. [4].

Starting from the idea that the basis of an active and healthy life is theoretical knowledge and understanding, Tittlbach et al. (2020) [30] created a program to implement theoretical knowledge in the seventh, eighth, ninth and tenth-grade students. For an entire school year, the experimental group was subjected to a program that was aimed at acquiring theoretical knowledge and understanding it. One of the peculiarities of the program was the fact that content planning was decided through a process involving scientists, students, school principals, and physical education teachers. In this format, rigorously organized meetings were scheduled in which it was decided that the intervention program should be based on the following four student-centred strategies: cognitive activation, reflection, correlation with daily life, collaborative learning. Cognitive activation involves students bringing examples of activities related to the learning unit to be taught. In this way, links are created between previous knowledge and new content. Reflection is achieved through various stimuli that students experience so as to help them in the process of reflection on the effects that physical activity has on the body (e.g., the effects of intense physical activity on heart rate or the effects of stretching on mental state). Correlation with daily life refers to the approach to content that is relevant to the student's daily life. Collaborative learning involves providing students with tasks that need to be solved in a team [30].

The danger posed by cardiovascular diseases is becoming more and more real, and the new generations need help in order to fight the growing number of cases. Lionis et al.

(1991) [31] attempted to do this in 13–14-year-old students by implementing a one-year theoretical course. The main objective of the course was to help students become aware of risk factors and resist external influences. The intervention program was taken over and adapted from the American version called "Know Your Body", and the main topics on which the content focused were: nutrition, physical activity and smoking. Another aspect, from the intervention of Lionis et al. (1991) [31], which we consider important to mention, is the fact that a textbook for students and a guide for teachers was used. In addition to these two, worksheets, videos, posters and "health passports" were also used (these were distributed at the beginning of the exam and each student had to record the results of the medical tests performed at the beginning and end of the course).

An innovative approach to intervention for seventh-grade students, was presented in the work of Frenn et al. (2005) [32]. Based on the transtheoretical model of Prochaska & DiClemente (1984) [33], the intervention program delivered theoretical content to students in a combined internet/video form. To begin with, the stage of change in the behaviour of each student was established, according to the transtheoretical model of Prochaska & DiClemente (1984) [33]. Then, adapted to stages 1 and 2 (pre-contemplation and contemplation), content was used that aimed at the process of awareness and self-evaluation of eating habits and physical activity. A computer-generated questionnaire was used for the "action" and "maintenance" stages, which automatically provided feedback based on the student's answers. In this way, children received feedback on their responses, which contained personalized information about what actions to take, or they received encouragement if their physical and eating behaviours were among the best. The results indicated that students who attended at least half of the program, recorded a decrease in the percentage of fat in the diet and an increase in the volume of physical activity [32].

Unlike studies that assess the volume of knowledge that students possess, the work of Placek et al. (2001) [16] focused on quality-that is, how this theoretical knowledge is perceived. So, interviews were organized with the students, in which their understanding related to a series of essential notions in the field of physical activity was analysed. The results revealed that, for the most part, the message perceived by 6th graders is that fitness is synonymous with physical appearance. For most students, this means being slim or, more precisely, in their terminology, "being skinny". Both boys and girls understood that being in good shape means being slim and looking good. After analysing students' perceptions of a wide range of concepts, the authors conclude that teachers should not only provide information to students, but they should also make sure that they understand them correctly because it has been shown that most of them retain these lifelong perceptions.

After the review of these interventions conducted at the middle school level, one may notice a change in the methods by which the theoretical content is delivered to students, compared to elementary school. This time, the characteristics of teaching theoretical content are more academic.

One may notice a difference in the topics addressed. At this educational level, the interventions offer knowledge that focuses on everyday life and its main problems. Students are encouraged to work in groups, to be as aware as possible of the main risk factors and the ways in which they can act most correctly.

Compared to the previous level, where knowledge was implemented through play and storytelling, this level shows a shift to everyday life and attempts at empowering students with directly applicable knowledge—such as designing a physical activity plan or calorie intake calculation.

### 3.3. High School (about 14–18 Years of Age)

In 1991, the project "Active Teen Project" (PAT) was initiated [34], which aimed at investigating the differences between students undergoing a conceptual physical education course and those who participated in traditional lessons, those based on sports. The main idea on which the research was built was the alarming figures that indicate sedentarism among young people. Four of the recommendations made by the National Centre for

Health Statistics (2001) [35] were used as benchmarks: those on a moderate level of physical activity (objective 1.3.), those on intense physical activity (objective 1.4.), those on sedentary lifestyle (objective 1.5) and those related to muscle toning and flexibility (objective 1.6.).

The content, based on the program "Fitness for Life", was offered to 1500 students in their first year of high school. Of the five lessons per week of physical education, two were assigned to the program, and the other three took place in the usual format. "Project Active Teen" used the two hours as follows: one was 100% theoretical, in which students used a textbook and were taught proven facts, basic concepts related to physical activity, ways to practise physical activity, goal setting and personalized program planning; and the second lesson took place in a gym similar to the one in which most adults today choose to work out. Practical lessons were held here in which the knowledge taught in the theoretical lessons was applied. Therefore, they focused on ways of self-assessment, necessary skills for building the exercise program and methods to practise a wide range of physical activities [34].

The results of the study highlighted that, after three years from the implementation of the program, students who participated in the conceptual physical education course were much more active at the end of high school than their colleagues, who participated in the traditional approach program. Another notable result was among female students: the probability of reporting sedentary behaviours was much lower in students who participated in the program [34].

The group of students who participated in the "Project Active Teen" (PAT) program [34] were tested in another study one year after high school [36]. The results of this study are all the more interesting as the students tested have now reached adulthood, and the results indicate a lower percentage of sedentary people in the group subjected to conceptual physical education (4% men and 10% women), compared to 21% (men and women), who participated in traditional physical education lessons [37].

Based on the students who participated in the "Project Active Teen" (PAT) program [34] a third study was conducted at a distance of 24 years from the date when they participated in the conceptual physical education course (20 years after high school) [38]. The results collected now from the group that participated in the program were compared with the results of a national study with similar people who participated in traditional physical education lessons—because it was not possible to make a comparison with the same control group from the original study. Following this third study, 24 years later, the authors reported that students who participated in the conceptual course of physical education have an optimal level of moderate physical activity better than the percentage of the national group who participated in traditional PE classes. As far as the second objective is concerned, that of muscle strengthening exercises, the differences are smaller, but there is still an advantage for the group that participated in the program, and regarding the third objective, that of a sedentary lifestyle, the values are also significant: 0% of men and only 5% of women who participated in the program fall into the sedentary category—these figures come in comparison with the percentages of the national group, where 25% of men and 27% of women fall into the sedentary typology [38].

Another result of this study, which we consider of great importance, is that out of the students who participated in the PAT project, 24 years apart from the conceptual course of physical education, 56% said they remember the information from the course, 50% said that they still use that information, 47% considered that the course was useful for them throughout their lives and 92% considered themselves well-informed in terms of physical activity and fitness [38].

The Curriculum Council of Western Australia (2009) [39] proposed a physical education course called "Physical Education Studies (PES)" for the 11th and 12th grades. The distinctive feature of this course, as it is described in the curriculum, is that the central element is represented by "achieving an integration of theory and practice" ([40], p. 2).

In another intervention, Stewart & Mitchell (2003) [41] conducted a study in which students were asked to design a fitness program for the development of a fitness component.

The results indicated that most students had problems applying the basic principles, and the areas where extremely low scores were recorded were the following: (1). the concept of specificity (selecting an appropriate activity to improve a fitness component); (2). goal setting (setting a specific, realistic goal to improve a fitness component over a period of time); (3). application of principles (understanding the relationships and interactions between principles-especially the concept of intensity).

Minana & Monfort (2020) [42] conducted a study on students aged 12 to 18, in which they analysed the level of knowledge regarding the prevention of back problems. The authors started this work from previous research, which states, on the one hand, that back pain is the most common problem among adults, and on the other hand, that the acquisition of knowledge about the lumbar/lower back area is the main tool for reducing these conditions. Finally, as a conclusion, the authors recommend that physical education teachers pay more attention to this topic when planning their content to be taught.

Interventions aimed at implementing theoretical knowledge at this age level seem to provide the clearest evidence that the knowledge-based approach has significant long-term results. This was also possible because at this age level, an attempt was made at implementing a theoretical component prior to the other levels, which allowed the observation of long-term results-such as the PAT project, where the results were studied 24 years after the intervention.

For many students, this level of education is the last chance for the education system to provide the necessary knowledge to help them plan and conduct the physical activity for health purposes. But, as shown by the study led by Stewart & Mitchell (2003) [41], students at this level have serious shortcomings regarding theoretical knowledge.

One may notice that the implementation of theoretical knowledge at the other educational levels represents steps that offer the possibility to offer some knowledge of the highest possible complexity at this level.

*3.4. The University Level*

After the end of the Second World War, most faculties included between 2 and 4 semesters of physical education courses to keep the population in good physical shape in case of another war [43]. Over time, these courses have undergone a number of changes, and the first dating of a theoretical physical education course was found in 1960 at the University of Illinois and Texas A&M University, where a special textbook was used, specially designed for this course [44,45]. This theoretical approach, together with the use of the textbook, represented an innovation in the field of physical education that was considered, until then, only in its practical aspect.

This new way of organizing PE lessons met with resistance to change, but in the 1970s and 1980s, there was an increase in the number of universities offering students a theoretical course in physical education [9]. Until 2009, the name of the course took no less than 58 forms, depending on each university, the most common names being: "Lifetime Fitness" (6%), "Health and Wellness" (4%), "Lifetime Wellness" (4%), "Fitness and Wellness" (4%) and "Fitness for Living" (4%) [44].

Until 1995, most colleges in the United States, regardless of their specialization, offered at least one physical education course as an integral part of general student education. In building this course, two approaches were differentiated. Thus, the universities chose one of the variants, or a combination of the two. The first type, of course, is characterized by physical activity—here students participate in physical education sessions that aim at developing motor skills. The second type of approach refers to a course in which the concepts underlying the ability to self-conduct physical activity are taught. Here, students participate in courses/seminars in which theoretical content is taught [46].

An analysis by Kulinna et al. (2009) [44] showed the evolution of theoretical physical education courses at the level of college students. Thus, if in 1990, 34% of the universities surveyed offered such a course [47], in 2000 there was a small decrease, to 33% [48] and in 2009, 44% of the universities surveyed replied that they offer a conceptual course in

physical education [39]. Further on, we shall present the contents of the theoretical course that we ordered according to the frequency of their reporting: "Benefits of Physical Activity; Cardiovascular Fitness; Nutrition; Lifestyles for Health, Wellness, and Fitness; Muscle Fitness: Basic Principles and Strength; Flexibility; Muscle Fitness: Muscular Endurance and General Muscle Fitness; Body Composition; Stress Management; How Much Physical Activity is Enough?; Choosing Nutritious Foods; Lifestyle Physical Activity and Positive Attitudes; Relaxation and Time Management; Cancer, Diabetes and Other Health Threats; Use and Abuse of Tobacco; Active Aerobics and Recreation; Use and Abuse of Alcohol; Learning Self-Management Skills; Safe and Smart Physical Activity; The Physical Activity Pyramid; Recognizing Quackery: Becoming an Informed Consumer; Physical Activity: Special Considerations; Toward Optimal Health; Making Informed Choices; Avoiding Destructive Behaviors; Active Sports and Skill-Related Physical Fitness 50 Body Mechanics; Making Consumer Choices." ([44], p. 129).

The impact of an optional physical education and health course in college was analyzed in a study by Pearman et al. (1997) [49]. The results showed that students who took such a course had more knowledge about blood pressure, blood cholesterol, and the recommended amount of fat in the diet compared to their colleagues who did not choose to participate. In addition to the benefits expressed in the acquired theoretical knowledge, it was also reported that the students who participated in the course had a positive attitude towards exercise, healthy eating and were against smoking. All these changes constituted factors that increased their chances to engage in physical activities and make beneficial changes in their diet. Students who attended the course were also less likely to start smoking. The findings of this study highlight the effects of an optional physical education and health course on the acquisition of knowledge and the building of favourable habits of college students.

Slava et al. (1984) [45] studied the effects that a theoretical course in physical education has on university students. Thus, the experimental group was subjected, for two years, to a course in which concepts related to physical activity were theoretically taught. The authors of the study built the profile of each student based on attitude, knowledge and degree of involvement in physical activities. The reported results highlighted that the students who participated in the theoretical course of physical education had profiles that did not resemble at all those of the control group. The most obvious differences were in terms of knowledge gained, but there were also improvements in attitudes and in the involvement in physical activities. In conclusion, this study demonstrated the long-term positive effects of a university course in which theoretical concepts related to physical activity are taught.

The same conclusion was reached by Maldari et al. (2021) [50], who noticed that, after 15 weeks in which students participated in the theoretical course, they experienced an increase in the level of moderate and intense physical activity. Considerable improvements were also made in terms of theoretical knowledge.

These physical education theoretical courses seem to have become the norm in universities in several countries, but especially in the USA. The popularity of these courses, at this level, could be explained by the fact that students feel much more strongly the need for a rational understanding of the concepts and principles that underlie physical activity at this age.

At the university level, the beginnings of this knowledge-based approach seem to be found. Here we find the earliest attempts at implementing theoretical content, and the USA seems to be the country that has implemented them the most at this level. This situation is also due to the fact that many American universities are involved in sports activities, and most of these universities offer optional physical education courses for students.

*3.5. Physical Education Teacher Education (PETE)*

From our analysis on how the theoretical component is implemented in students of different educational levels, research on how future physical education teachers are prepared to implement this component could not miss. Garrett & Wrench (2007) [51] warn

that students who are preparing to become physical education teachers will not react the same way even if they are subjected to the same training program. The authors conducted research that emphasizes that each student comes up with their own concepts about sports and physical activity-which are formed as a result of their own experiences. Therefore, the teacher training process cannot be viewed in a simplistic manner. Dowling (2011) [52] considers that, when physical education teachers are trained, we must ask ourselves, "what concepts of teacher professionalism do we offer students explicitly and implicitly, intentionally and unintentionally, via our words, our actions and/or our inactions?" [52].

Santiago & Morow (2020) [53] highlighted the fact that students who are preparing to become physical education teachers have a low level of knowledge aimed at fitness guidelines for optimal health. These results are in agreement with Bulger et al. (2001) [54], who consider that the programs that train future physical education teachers do not provide them with the necessary training to respond to the current responsibility-that of developing skills, knowledge, attitudes, and an optimal level of motor ability for an active life in students. Barnett & Merriman (1994) [55] also reported an average level of less than 80% of potential physical education teachers on a knowledge test about fitness. Petersen et al. (2003) [56] also reported knowledge gaps of future specialized teachers, with average results of 75.2% recorded in a knowledge test of low difficulty. (FitSmart) Miller and Housner (1998) [57] measured the level of fitness knowledge for the health of both students preparing to become teachers and that of active teachers. The recorded results led the authors to state that the level of knowledge of both groups (i.e., active students and teachers) is unsatisfactory as far as the ability to teach such theoretical content to students is concerned.

Universities such as Arizona State University, Appalachian State University, or The Ohio State University have identified the need to prepare future teachers for the requirements of today's society. For this, they decided to offer, in their programs, a course in which to equip students with the necessary techniques for teaching theoretical knowledge to complement the practical activity in the physical education lesson. Here, students are prepared how to teach a theoretical component in the classroom, using a textbook and other helpful materials [13].

Here are some of the topics addressed by these conceptual physical education courses within PETE programs: "PE Program Philosophy and Theoretical Foundations; Physical Education Content Standards, FE Benchmarks, and Physical Literacy Overview; Overview of CPE/FE Content Knowledge (e.g., common content, specialized content); Methods of Presenting Classroom Content (e.g., use of AV, classroom discussions); Methods of Presenting CPE/FE Activity Session Content; Overview of Available Programs (e.g., Fitness for Life, Physical Best); Overview of Student and Program Evaluation (e.g., tests, portfolios, projects, workbooks); Integrating CPE/FE with Other Physical Education Programs (e.g., Sport Education, Traditional PE); Integrating CPE/FE with Whole-of School Programs (e.g., CSPAP, PYFP); Using the Web and e-books in CPE; Online CPE: Pros and Cons, Guidelines for Implementation; "([13], p. 48).

A program for physical education teachers this time is "Promoting Active Lifestyles (PAL)" [58]. The purpose of the program is to challenge teachers to re-evaluate the philosophy on which their entire pedagogical activity is based on the discipline of physical education. To do this, during the program (2016–2017), teachers were introduced to a series of paradoxes on which they were invited to reflect:

- Promoting an active lifestyle is usually evident in the philosophy of physical education, but it is not as obvious in the discipline curriculum. For example, while teachers say they encourage and educate students to engage in physical activity throughout their lives, their planning materials do not reflect this.
- Physical education lessons provide an opportunity for students to be physically active. However, the level of physical activity in the lessons is low. For example, even if students regularly participate in physical education lessons, it seems that they cannot be considered active during the lesson.

- Physical education teachers usually say that they use motor performance assessment to promote physical activity. However, many students hate it and, in addition, they do not learn anything from the assessment activity. For example, motor assessment is usually included in the curriculum to encourage students to be physically active, but many of them do not feel any pleasure when they participate in it and gain very little knowledge and understanding from it.
- Physical education teachers help students gain knowledge and understanding so that they can lead an active life independently. Nevertheless, many students are confused about this topic. For example, students have the opportunity, during the physical education lesson, to gain an understanding of physical activity, but many of them have misunderstandings of the subject-such as thin people are healthy or one needs to run fast to be healthy ([58], p.3).

After the teachers were invited to analyse, reflect and discuss these paradoxes, their conclusions were cumulated into three major themes: (1) Teachers described the contact with the PAL paradoxes as interesting, surprising and complicated; (2) Teachers have expressed a burning desire to discuss and solve these paradoxes. (3) Teachers experienced a sense of pleasure but also a challenging one regarding the possibility to influence the philosophy and pedagogical approach of other colleagues ([58], p.3).

It is obvious that, at this level, the knowledge-based approach has a solid foundation. All experiments and studies conducted by researchers aim at producing changes at this level. Only if the future physical education teachers are prepared to teach a theoretical component along with the practical one, can we expect a real change.

The transition from a sport-based approach to a knowledge-based approach in PE largely depends on changes in the training of future teachers. Future teachers need to be trained regarding the tools and methods necessary for teaching the theoretical component. It is also necessary for the pedagogical practice to contain such lessons focused on teaching theoretical notions.

From the information presented, one may rightly draw the conclusion that America is the country that has brought real changes in the training of future teachers. The same country is the one that has implemented clear theoretical objectives for each grade (Standard 4) in the teaching of PE, within public education.

*3.6. Tools for Assessing the Cognitive Component in Physical Education*

In implementing the cognitive component in PE, it is not enough to identify the content to be taught and understood-assessing the level of assimilation and understanding is equally important [59]. For this fact, we will continue to try to present the main tools used to assess the cognitive component of the modern approach to physical education, which is called physical literacy. Dawn et al. (2009) [60] state that the traditional form of assessment in PE is focused either on elements of fitness or an out-of-context approach is preferred, as is the case with the assessment of individual motor skills. In contrast to this view, Cale & Harris (2018) [59] propose an assessment of the theoretical knowledge gained by the students and their understanding through written, oral tests, active answers, or observation. Morrow et al. (2011) [61], in the book "Measurement and Evaluation in Human Performance," state that "evaluation in physical education also includes the essay, presentation, portfolio and tests with complex answers" ([61], p. 381).

After studying the domain-specific literature, we grouped the tools for evaluating the cognitive component into three categories: questionnaires, interviews, interactive methods.

3.6.1. The Questionnaire Is Undoubtedly the Most Obvious Method for Assessing the Cognitive Component of Physical Literacy

Therefore, we found the method of the questionnaire in several intervention programs, as a tool for measuring the volume of knowledge acquired or for measuring the degree of understanding.

As we have seen, the cognitive component of physical literacy can encompass a wide range of fields of knowledge: from physiology, biomechanics or anatomy to psychology or even philosophy. Therefore, in each of the intervention programs identified by us, questionnaires adapted to the content taught were used. In Table 2, we provided a description of the objectives stated in drawing up the main questionnaires used in cognitive interventions in PE classes.

**Table 2.** Questionnaires that assess the cognitive component of PL.

| The Questionnaire | Assessed Components |
|---|---|
| Longmuir et al. (2018), [62] | (1) Physical activity (how to move); (2) interpretation (movement assessment); (3) health and fitness (the importance of exercise, need for relaxation and sleep, etc.); |
| Sorensen et al. (2013), [63]-European Health Literacy Survey Questionnaire (HLS-EU-Q) | (1) A health literacy section; (2) a section on the determinant factors and outcomes associated with health literacy; |
| Zhu et al. (1999), [64]-FitSmart | The contents of Standard 4, which focuses on the knowledge of physical fitness concepts, principles, and strategies to improve health and performance (for students between 12–18 years old in the USA) |
| Society of Health and Physical Education (2010), [65]-PE Metrics | The contents of Standard 4, which focuses on the knowledge of physical fitness concepts, principles, and strategies to improve health and performance (for students between 5–10 years old in the USA) |
| Minana & Monfort (2020), [42] | (1) Exercises specific to the back area; (2) lumbar problems; |
| Johnson-Taylor& Everhart (2006), [36] | examination of healthy eating habits; |
| Turnin et al. (2001), [66] | (1) Habits related to the food consumed; (2) eating habits; (3) nutritional knowledge; |
| Calfas (1991), [67] | nutritional knowledge; |
| Lionis et al. (1991), [31] | (1) General knowledge about health; (2) smoking; (3) nutrition; (4) blood pressure; (5) physical activity; (6) dental health; |
| Caballero (2003), [24] | (1) Physical activity; (2) diet; (3) attitude and behaviour towards body weight; (4) cultural identity; |
| Sun et al. (2012), [20] | (1) Specialized vocabulary; (2) scientific principles of involvement in physical activity; (3) measurements of the physiological response to physical activity; (4) the implications of exercise intensity; (5) the physiological benefits of physical activity; |
| Resnicow et al. (1993), [28] | the level of theoretical knowledge, the attitudes and behaviours regarding a healthy life; |
| Slava et al. (1984), [45] | (1) Attitude towards physical activity; (2) the knowledge that the student possesses about physical activity; (3) data about the physical activity they perform; |
| Dale et al. (1998), [34] | (1) The level of involvement in moderate and/or intense physical activity; (2) The level of involvement in activities that involve the development of muscle strength and elasticity; |

A questionnaire that we chose to present separately is the one proposed by Fren et al. (2005) [32]. This questionnaire was developed based on the transtheoretical model of behavioral change developed by Prochaska & DiClemente (1984), [33], and the distinctive feature is that it provides real-time feedback based on the student's answer in the form of suggestions to help the student overcome the stage they are at.

3.6.2. The Interview-Is the Tool That Allows a Greater Mobility of the Respondent's Answer

In the study carried out by Placek et al. (2001) [16], the authors were not interested in measuring the level of participation in physical activities and the volume of theoretical knowledge related to PE. The main objective was to provide a more detailed picture of the

concepts that sixth-grade students have about fitness. For this, the students were asked if and why people should do physical activity and then, they were asked to freely define, with a complex answer, the terms used in fitness. They were also asked to make activity suggestions for a friend who would like to get in good shape. They were also shown drawings of people participating in various physical activities (e.g., cycling, weightlifting, stretching, etc.) and were asked what component of fitness is developed. Finally, interviews were organized with the students, during which they were asked to clarify the answers and say where they got the knowledge. The interviews were transcribed verbatim and then divided into categories according to the ideas related. For the analysis of the results, the comparative method of qualitative analysis developed by Glaser & Strauss (1967), [68] was used.

### 3.6.3. Interactive Methods Are Assessment Tools Developed in Order to Provide Students with an Attractive Method of Assessment

Morrow et al. (2011) [61] note that the written test may be unpleasant for students, so they propose two much more attractive approaches: creating a magazine and a portfolio. To create a magazine, the authors propose the formation of groups of six students in which each is responsible for writing an article on the topic of the magazine. This topic could be about a sports branch, physical activity, opportunities offered by the community to exercise, children's health, etc. The second type of attractive evaluation proposed by Morrow et al. (2011) [61] of the theoretical component of physical education is the portfolio. "The portfolio provides documentation on students' learning, knowledge and skills that the teacher wants each student to have documented" ([61], p. 387).

The Curriculum Council of Western Australia (2005) [39] proposed for the 11th and 12th grades a course called "Physical Education Studies". What is special about this course is that it focuses more on the theoretical side of PE than on the practical one. This fact results from the fact that the final grade is made up of 70% (the theoretical evaluation) and 30% (the practical one). For the theoretical part, students must, among other things, present the results of a research activity that involves planning, conducting and communicating the results of an investigation regarding their own participation in physical activity. This involves: the potential for participation, problems in physical activity and the social context. The research results can be communicated in any form: written, oral, graphic, video, or combinations thereof [69]. Sun et al. (2012) [20] conducted a similar intervention program in which the evaluation took an innovative form. Students in this program were assessed on the basis of diaries they had to keep in which to make predictions, write down observations and collect data on how their body reacts to physical activity.

Active assessment is another interactive method that involves active/practical answers to questions and/or requirements that involve proving the acquisition and understanding of the theoretical content regarding PE. Some examples of this evaluation type are given in Cale & Harris's paper (2018) [59]: (a) Name and show the muscle that works the most when you run/jump/throw! (b) Why is it important to stretch your muscles after working out hard? (c) Explain to a colleague why physical activity is good for your health. (d) Which are the main reasons why people are not physically active? (e) Show a muscle-stretching exercise [59].

## 4. Conclusions

The cognitive component of physical literacy is the key element that transforms the physical education lesson into a meaningful one for the student. This component is approached by a growing number of researchers, and this was also noticed by Demetriou & Höner (2012) [70], who conducted a review of the interventions carried out in schools with a physical activity component.

The mix between theoretical knowledge and their practical experimentation seems to be the combination that comes closest to the educational character of PE. And this conclusion is in agreement with De Bourdeaudhuij et al. (2011) [71], who conducted a review of school-based interventions promoting both physical activity and healthy eating

in Europe. They suggest that combining educational and environmental components have better and more relevant effects.

This new "knowledge-based" approach is one that can be implemented at any age, as long as the methods are adapted. The advantages of this approach have been demonstrated in numerous studies, some of them carried out over significant study periods (24 years since implementation).

Another extremely important advantage is related to the students' grading. The student's level of theoretical knowledge can be a grading criterion, and thus solves an important problem of the "sport-based" approach, i.e., students are graded only on the basis of physical performance, which is an injustice, because students have biological endowments that are much different from each other. From this perspective, our paper presents a wide range of tools used in the evaluation of the theoretical component, some of them being very attractive for students.

Thus, we may conclude that the knowledge-based approach is more than an isolated attempt at changing PE, it is a reality that has proven its effectiveness and applicability at all levels. In addition, it has also been implemented under various names in educational systems in the USA, Australia and several Western European countries.

**Author Contributions:** All authors have an equal contribution to the publication. All authors have read and agreed to the published version of the manuscript.

**Funding:** This research received no external funding.

**Institutional Review Board Statement:** Not applicable.

**Informed Consent Statement:** Not applicable.

**Data Availability Statement:** Not applicable.

**Conflicts of Interest:** The authors declare no conflict of interest.

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
