# Peer review of "Interventions Which Aim at Implementing the Knowledge-Based Approach in the PE Lesson: A Systematic Review"

_sustainability, doi:10.3390/su132111781_

Round 1

Reviewer 1 Report

This paper has focused on the Interventions implementing the cognitive component from Physical Literacy in the PE lessons. The topic is timely and potentially important. However, the manuscript needs some revisions:

The abstract is not very clear. An abstract is a complete version or form of your article. This is the portion of the manuscript where the authors provide a summary that presents the manuscript's most important features. 

The introduction is too vague on what you intend to do and why it matters. Also, you need to be very clear on why this topic is important and how your study, a systematic review, has contributed to the field. In other words, given how little is known, there needs to be a stronger argument for researching the gap in this area.

The section Materials and Methods needs some more work. For example, you may need to be more specific on your method description rather than simply stating “Because we want to highlight the ways in which the cognitive component can be implemented in PE, we will carry out a qualitative analysis of the identified works.” Secondly, you need to provide some rationales on why you selected SCILIT database; how this single database can sufficiently support your study which is supposed to be a systematic review.

In addition to the key words, did you have any other article selection/exclusion criteria? In summary, we need to know whether the studies you have selected are representative enough for your review. This is the key to the whole systematic review.

About the discussion and conclusion, it is acceptable to present/discuss your findings by age levels. However, you need to integrate what you have and provide a more in-depth and informative discussion. For example, I was trying to read some discussion on the difference and similarities of those interventions across different levels and why. Also, Physical Education Teacher Education (PETE) seems not fit into this “age level” design.  If it is necessary, you may need to provide a bit more explanation why it’s there.

By the way, a thorough editing is needed for this paper. For example, clearly the paper is written by multiple authors. However, I saw “I” in the abstract. A careful proofreading would have helped you to avoid the mistake like this.

Reviewer 2 Report

Thank you for the opportunity to review your manuscript. I hope that you find my comments and edits attached helpful has your being the process of revision.  

  1. Abstract: There is a big part for background and the data analysis description is very short (authors use the same sentence that in the text). Also in abstract, there is a mistake. They say "For this research I used...", but there are several authors. Besides the abstract, the same sentence is wrong in page 4.
  2. Materials and Methods: Authors have to explain more the journal selection criteria as dates (which years were investigated? and why?) or language (Just English? Also Russian? Other languages?). Which countries implemented the 31 works they studied?
  3. Also, it could be more clear if they change the sentence "the characteristics of the intervention and on the other hand, the tool used to..." Instead of that one, it could be better to say "the characteristics of the interventions, grouped according to the the ages of the groups the are for, and on the hand, the tool used to...."
  4. Also, authors say "Afterwars, we extracted...", but it is not "afterwards". The analysis process is based on this criteria for categorization...
  5. In page 6, authors show the first section "The pre-pubertal period (6-10 / 11 years of age). But I ask, Why they didn´t study any work for kindergarten? It is strange that authors studied the standar progression from kindergarten to 8th grade, and later they didn´t find any work for this ages.
  6. In the section 3.4 "The university level", authors hace to concrete the countries they are investigating (just USA and Australia?) because in several countries of Europe it is not usual to have PE at the University (just fo specific Grades, like future teachers).
  7. Table 3 has to be fix. In the first line, it is missed a comma before "etc."
  8. Table 3, in the 3rd (Zhu el all, 1999) and 4th lines (Society of Health and Physical Education, 2010), it should be concreted the issues that were assessed, like in the other lines.
  9. Table 3, lines 12 (Resnicow et al. 1993), it is not necessary to repeat "within the Know your body program, standardized questionnaries adapted for each class are developed. These questionnaries aimed to assessing". Authors can say "the level of ...", as it is said in the rest of the table, just the issues that are assessed.
  10. The first sentence after table 3 has a mistake. It has to say "Fren et all (2005). But is says "Fren et. All (2005)".
  11. The manuscript is based on the review of literature. The selection of the papers is not done very sistematically. If authors would use other database, they would complete the study properly (for example, they missed studies for kindergarten).

Round 2

Reviewer 1 Report

The authors' responses and revisions are acceptable.